# Role of STIM1 in the Regulation of Cardiac Energy Substrate Preference

**DOI:** 10.3390/ijms241713188

**Published:** 2023-08-25

**Authors:** Panpan Liu, Zhuli Yang, Youjun Wang, Aomin Sun

**Affiliations:** 1Beijing Key Laboratory of Gene Resource and Molecular Development, College of Life Sciences, Beijing Normal University, Beijing 100875, China; 2Key Laboratory of Cell Proliferation and Regulation Biology, Ministry of Education, College of Life Sciences, Beijing Normal University, Beijing 100875, China

**Keywords:** STIM1, cardiac energy metabolism, cardiac hypertrophy, diabetic cardiomyopathy, glucose, fatty acid

## Abstract

The heart requires a variety of energy substrates to maintain proper contractile function. Glucose and long-chain fatty acids (FA) are the major cardiac metabolic substrates under physiological conditions. Upon stress, a shift of cardiac substrate preference toward either glucose or FA is associated with cardiac diseases. For example, in pressure-overloaded hypertrophic hearts, there is a long-lasting substrate shift toward glucose, while in hearts with diabetic cardiomyopathy, the fuel is switched toward FA. Stromal interaction molecule 1 (STIM1), a well-established calcium (Ca^2+^) sensor of endoplasmic reticulum (ER) Ca^2+^ store, is increasingly recognized as a critical player in mediating both cardiac hypertrophy and diabetic cardiomyopathy. However, the cause–effect relationship between STIM1 and glucose/FA metabolism and the possible mechanisms by which STIM1 is involved in these cardiac metabolic diseases are poorly understood. In this review, we first discussed STIM1-dependent signaling in cardiomyocytes and metabolic changes in cardiac hypertrophy and diabetic cardiomyopathy. Second, we provided examples of the involvement of STIM1 in energy metabolism to discuss the emerging role of STIM1 in the regulation of energy substrate preference in metabolic cardiac diseases and speculated the corresponding underlying molecular mechanisms of the crosstalk between STIM1 and cardiac energy substrate preference. Finally, we briefly discussed and presented future perspectives on the possibility of targeting STIM1 to rescue cardiac metabolic diseases. Taken together, STIM1 emerges as a key player in regulating cardiac energy substrate preference, and revealing the underlying molecular mechanisms by which STIM1 mediates cardiac energy metabolism could be helpful to find novel targets to prevent or treat cardiac metabolic diseases.

## 1. Introduction

To maintain an optimal contractile function, the heart has a high demand for energy substrates to continuously form energy-rich phosphate bonds (i.e., intracellular adenosine triphosphate, ATP) [1]. Multiple substrates can be utilized by the heart to produce the necessary ATP, including glucose, lactate, long-chain fatty acids (FA), amino acids and ketone bodies, among which glucose and FA are the major metabolic substrates to generate acetyl-CoA for the tricarboxylic acid cycle and subsequent mitochondrial oxidation in healthy heart [2]. There is a balance between glucose and FA utilization in a healthy heart: FA contributes 60% of the ATP production via β-oxidation, while glucose provides 30% through glycolysis and glucose oxidation [3] (Figure 1). The remaining 10% of ATP is produced by lactate with minor contributions from ketone bodies and amino acids, which will not be addressed in this review. However, this kind of balanced utilization of substrates can be changed toward a preference of either glucose or FA upon different pathological conditions (e.g., cardiac hypertrophy and diabetic cardiomyopathy), which causes gluco/lipo-toxicity and eventually cardiac dysfunction [4,5,6,7] (Figure 1).

Stromal interaction molecule (STIM) proteins are endoplasmic reticulum/sarcoplasmic reticulum (ER/SR) Ca^2+^ sensors. There are two known members of STIM in mammalian cells: STIM1 and STIM2. STIM1 is the major mediator of the classic store-operated calcium entry (SOCE), while STIM2 works more as a coordinator and regulator of STIM1 [8,9,10,11,12] and as a rheostat for basal cytosolic Ca^2+^ concentrations [13]. Sensing ER/SR Ca^2+^ store depletion, STIM1 and STIM2 become activated and engage with the pore-forming Orai1 ion channel proteins, localized in the plasma membrane, thereby triggering channel opening and activating the previously characterized I_CRAC_ (Ca^2+^ release activated currents) [14]. As most investigators use Ca^2+^ imaging techniques to report STIM-Orai1 activity, this process is usually termed Store-Operated Calcium Entry (SOCE) [8,9,10]. The concept of SOCE was first introduced by Putney in 1986 [15], and currently, its role in non-excitable cells is well established. By comparison, despite a number of reports showing the function of STIM1 in the heart [16,17,18,19,20,21,22,23], the physiological and pathological roles of STIM1 in cardiomyocytes remain largely elusive. The STIM1 expression level in cardiomyocytes is dependent on the developmental and pathological states of the heart. STIM1 is highly expressed in neonatal cardiomyocytes but not adult cardiomyocytes [24,25,26]. It is believed that STIM1-mediated SOCE is important for neonatal heart growth via the activation of the calcineurin/nuclear factor of activated T cells (NFAT) signaling pathway [27]. In addition, STIM1 is present in coronary sinus cardiomyocytes in a tract from the sinoatrial node to the coronary sinus, and STIM1-mediated SOCE is important for interatrial conduction and the maintenance of the normal heart rhythm [25,26,28]. Under the hypertrophic condition, the expression of STIM1 is robustly increased (Figure 1), and the genetic deletion or pharmacological inhibition of STIM1 decreased SOCE and protected the heart from hypertrophy [17,21]. Of note, the development of cardiac hypertrophy is associated with metabolic changes in cardiomyocytes [29]. Interestingly, the expression of STIM1 has been shown to decrease in diabetic cardiomyopathy [30,31] (Figure 1), and the cardiomyocyte-restricted deletion of STIM1 causes the same characteristics of metabolic changes as appeared in diabetic cardiomyocytes [32,33,34,35,36], which prompts us to consider the relevance of STIM1 in the regulation of cardiac energy substrate switch. In addition, STIM2 may also have an important role in guiding metabolism [37], especially as it regulates basal Ca^2+^ and STIM1 activation. Future investigations are needed to elucidate this.

In this review, we aimed to summarize the currently known roles of STIM1 in the energy metabolism of many types of cells with a focus on cardiomyocytes. We also gave some future perspectives of the possibility of targeting STIM1 to rescue cardiac metabolic diseases.

## 2. STIM1-Dependent Signaling

STIM1 is a single transmembrane protein resident in ER/SR. It consists of a signal peptide (SP), the EF-hand and sterile alpha motif (EF-SAM) domain in the N-terminus (in the lumen of ER/SR), coiled-coil regions, an inhibitory domain (ID) and a polybasic region (K) in the C-terminus (in cytosol) (Figure 2a). The EF-SAM domain is responsible for its Ca^2+^-sensing role. There are three coiled-coil regions: coiled coil 1, 2, and 3 (CC1, 2, 3). CC1 has an inhibitory effect on CC2–CC3, which is the minimum domain of STIM1 to activate Orai1 (STIM1-Orai1 activating region, SOAR) [10,38,39,40] (Figure 2a). Under the resting state, Ca^2+^ binds to the EF-SAM domain to keep this ER-luminal region monomeric; CC1 interacts with SOAR to maintain cytosolic STIM1 domains in an auto-inhibitory dimeric folding form [39,41,42,43]. When the ER/SR Ca^2+^ store is depleted, the decalcified EF-SAM domains dimerize to pull the TM domains closer, triggering CC1 to release SOAR and unfold the cytosolic region of STIM1, promoting its oligomerization and the activation of Orai1, allowing Ca^2+^ influxes, or SOCE [39,44] (Figure 3).

There are at least three known alternative splicing variants of STIM1 (STIM1L, STIM1A/β, and STIM1B) (Figure 2a). STIM1A/β has an insertion of 31-amino-acid peptide after the SOAR in the cytoplasmic domain, which is found in astrocytes, heart, kidney, and testes [45]. While STIM1A/β has been shown to more efficiently mediate SOCE with faster kinetics by disordering the cytosolic inhibitory domain in HEK, HeLa and glioblastoma cells [46], it was also shown to reduce SOCE in astrocytes in heterologous expression while at the same time increasing NFAT translocation due to a more efficient recruitment of the NFAT signalosome [45]. Indeed, it is possible that differential NFAT activation and not small differences in SOCE mediate a splice variant-specific effect in cells or else that the functions of STIM1A/β may differ in different cell types. STIM1B is a short isoform of STIM1, which lacks 170 amino acids but has an extra peptide of 26 amino acids in the cytoplasmic domain. STIM1B is a neuron-specific variant and induces slower I_CRAC_ and the inactivation of I_CRAC_ [47]. The C-terminal end of STIM1L bears an extra 106-amino-acids-long peptide that contains an actin-binding domain (ABD); thus, it can facilitate rapid SOCE [48,49]. STIM1L is only expressed in skeletal muscle cells, neonatal cardiomyocytes or hypertrophic adult cardiomyocytes [24].

In non-excitable cells, STIM1 is distributed throughout the ER at rest. Upon store depletion, STIM1 translocates to ER–PM junctions to interact and activate Orai1 to induce SOCE [10] (Figure 3). In cardiomyocytes, STIM1 is mostly localized in the SR at the Z-lines and is believed to function as a sensor for SR store [16,17,19,21]; the underlying mechanisms for this uneven distribution still awaits further investigation. Orai1 is detected in the sarcolemma membranes (Figure 2b). Around the area where STIM1 and Orai1 are resident is the well-known diad, which is a key place for excitation–contraction coupling (ECC) mediated by ryanodine receptor 2 (RyR2) at the terminal cisternae of the SR and voltage-gated Ca^2+^ channel Ca_v_2.1 at the T-tubule (Figure 2b). Additionally, SR Ca^2+^ ATPase (SERCA) and inositol-1,4,5-triphosphate receptor (IP_3_R) are also involved in Ca^2+^ handling in this area (Figure 2b) [50]. Different from non-excitable cells, STIM1 forms constitutive puncta near sarcolemma and does not change its distribution upon store depletion [25]. In addition, there is barely co-localization between STIM1 and Orai1 in cardiomyocytes [25], indicating fewer involvements of Orai1 in STIM1-mediated SOCE. Indeed, STIM1 can hardly induce classic SOCE with Orai1, which is characterized by highly calcium-selective I_CRAC_ with large inward rectification and very positive reversal potential [25,51]. Instead, a non-selective current is more commonly induced by STIM1 likely via the activation of transient receptor potential channels such as TRPC [51,52,53]. However, it is still controversial whether TRPC channels can be directly activated by STIM1 [54]. And Orai2 and Orai3 are both expressed in cardiac cells [23], thus it is also likely that STIM1 might mediate SOCE via interactions with Orai2 or Orai3.

STIM1-mediated SOCE is crucial for refilling ER Ca^2+^ [10,55,56] (Figure 3), as STIM1 deletion blocked the fast refilling of ER Ca^2+^ after store depletion [57,58]. Also, a decrease in STIM1 expression impaired ER Ca^2+^ refilling, which could be restored by STIM1 overexpression [30]. Similarly, it is believed that one key function of STIM1-mediated SOCE in cardiomyocytes is to refill SR Ca^2+^ and thereby maintain SR Ca^2+^ homeostasis for proper protein folding and processing [59]. STIM1 knockdown by siRNA in cultured neonatal rat ventricular cardiomyocytes reduced the SR Ca^2+^ content [16]. In addition, it has been shown that STIM1 overexpression increased the SR Ca^2+^ level in rat ventricular myocytes [52]. However, a study from Correll and colleagues showed that STIM1 overexpression had no effect on the total SR Ca^2+^ load in mouse ventricular myocytes. They suggested that elevated STIM1 resulted in increased Ca^2+^ uptake into the SR but also RyR2-dependent Ca^2+^ leak; therefore, the total SR Ca^2+^ load remained unaltered [19]. Ca^2+^ influx through STIM1-mediated SOCE refills ER/SR Ca^2+^ by SERCA. SERCA pumps have been shown to co-localize with STIM1 in different cell types, coupling Ca^2+^ entry with Ca^2+^ refilling [56]. In cardiomyocytes, SERCA is mainly resident at Z-lines where STIM1 is localized [60] (Figure 2b), likely enabling the similar function of Ca^2+^ entry–Ca^2+^ refilling coupling. A disruption of ER Ca^2+^ homeostasis can cause ER stress, leading to the accumulation of unfolded proteins and thus unfolded protein response (UPR) [61]. STIM1-knockout hearts displayed increased levels of ER stress marker CHOP [59]. Single-nucleotide polymorphisms (SNPs) in the STIM1 gene are correlated with ER stress in patients undergoing cardiac catheterization and spontaneous hypertension in rats [62,63]. Since ER stress and UPR often correlate with metabolic disorders [64,65,66,67,68], it is likely that STIM1 may mediate an energy substrate preference in cardiac metabolic diseases through regulating ER Ca^2+^ homeostasis.

In addition to refilling SR Ca^2+^, STIM1 could also activate downstream Ca^2+^-dependent metabolic pathways (Figure 3). It is widely accepted that Ca^2+^ signaling regulates cardiac energy metabolism through the direct activation of Ca^2+^-dependent proteins, kinases, enzymes and transcriptional regulators [69,70]. For instance, pyruvate dehydrogenase (PDH), PDH kinase (PDK), AMP-activated protein kinase (AMPK) and AKT are involved in the regulation of cardiac energy metabolism and are mediated by Ca^2+^ [71,72,73]. In addition, Ca^2+^ signaling regulates glycolysis and glucose oxidation in the heart and is crucial for the translocation of glucose transporter (GLUT) 4 and FA transporter CD36 (also known as the scavenger receptor B2, SR-B2) to the sarcolemma to induce cardiac glucose and FA uptake, respectively [74,75]. Moreover, peroxisome proliferator-activated receptor (PPAR) and PPARγ coactivator-1α (PGC-1α), key transcriptional regulators of expression of genes encoding mitochondrial oxidation enzymes, are associated with Ca^2+^ signaling as well [76,77]. Considering the key role of STIM1 in SOCE, STIM1 may regulate the energy substrate preference in cardiac metabolic diseases through Ca^2+^ signaling. Moreover, STIM1 can directly interact with a variety of other proteins [52,57,78,79,80,81,82,83,84], and alternative splicing variants of STIM1 might have splice-specific partners [45]. Interestingly, a recent report has shown that STIM1 negatively regulates the activation of stimulator of interferon genes (STING) through tethering STING to ER [85]. Of note, the cyclic GMP–AMP synthase (gGAS)–STING signaling pathway mediates cardiac metabolic abnormalities [86]. Therefore, STIM1 may regulate the energy substrate preference in cardiac metabolic diseases through the gGAS–STING signaling pathway or similarly via direct interactions with some other proteins correlating to energy metabolism. Given that glycolytic enzymes are clustered near SR [87] where STIM1 is resident, the likelihood of the conjecture increases.

Moreover, STIM1-mediated SOCE could contribute to cardiac metabolic diseases via the alteration of mitochondrial Ca^2+^ homeostasis or fission. Mitochondrial function including ATP production and reactive oxygen species (ROS) generation is regulated by Ca^2+^ [88,89]. Mitochondria takes up Ca^2+^ via voltage-dependent anion channel (VDAC) and mitochondrial Ca^2+^ uniporter (MCU) [90,91,92,93]. Normally, the ER/SR transmits Ca^2+^ to mitochondria to regulate mitochondrial function [94]. Therefore, STIM1-mediated SOCE is related to mitochondrial function through involvement in mitochondrial Ca^2+^ uptake from the ER/SR [19,95,96,97,98] (Figure 3). Alterations of ER/SR Ca^2+^ homeostasis due to the changes of STIM1-mediated SOCE would affect ER/SR–mitochondrial Ca^2+^ communication and thereby cause mitochondrial dysfunction. Excessive or insufficient mitochondrial Ca^2+^ in cardiomyocytes leads to mitochondrial dysfunction [99], which is associated with cardiac metabolic diseases such as cardiac hypertrophy and diabetic cardiomyopathy [99,100,101,102]. Interestingly, research has shown a decrease in mitochondrial Ca^2+^ uptake in diabetic hearts and restoring mitochondrial Ca^2+^ rescued mitochondrial and cardiac dysfunction [103]. In addition, Ca^2+^–calcineurin is involved in the regulation of mitochondrial fission, which is associated with the development of both cardiac hypertrophy and diabetic cardiomyopathy [98,104,105,106,107,108,109,110]. A recent study showed that STIM1 deficiency in cardiomyocytes changed mitochondrial morphology, which is indicative of an elevation of mitochondrial fission [32,59]. This phenomenon is in line with the fact that pro-fusion proteins (optic atrophy 1, Opa1; mitofusion 2, Mfn2) are downregulated, and the pro-fission protein (dynamin-related protein 1, Drp1) is activated, promoting mitochondrial fission in diabetic cardiomyocytes [98,108,109,110]. Notably, Mfn2 positively regulates SOCE via mediating STIM1 movement to ER–PM junctions [111], indicating there is a feedback loop between mitochondrial dynamics and STIM1-mediated SOCE. Therefore, any increase or decrease in STIM1 could induce mitochondrial dysfunction via influencing the mitochondrial Ca^2+^ level and mitochondrial fission, resulting in the alterations in cardiac glucose and FA oxidation. In turn, increased ROS due to mitochondrial dysfunction could downregulate STIM1 expression through NF-kB as a feedback effect [112]. Moreover, a study of T cells has shown that STIM1 deletion resulted in the downregulated expression of many subunits of the electron transport chain (ETC), suggesting that STIM1 is involved in mitochondrial function by controlling the expression of subunits of the ETC [113]. It would be intriguing to test this also the case in cardiomyocytes.

## 3. Glucose Preference in Cardiac Hypertrophy and FA Preference in Diabetic Cardiomyopathy

During the development of cardiac hypertrophy, there is a shift of energy substrate preference toward glucose utilization, which is caused by the altered expression and activity of transcriptional proteins related to glycolysis [2,114,115,116]. However, there is no parallel increase in glucose oxidation, which could be caused by an uncoupling between glycolysis and glucose oxidation [117,118,119,120]. In addition, hypertrophy also leads to hypoxia because of the increased diffusion distance of oxygen due to increased wall thickness [121]. Given that ATP production from glucose requires less oxygen than FA, it results in further glucose utilization as the main source of energy to generate ATP in hypertrophic heart [122]. In the end, cardiac energy supply cannot match energy demand anymore, which results in severe contractile dysfunction and eventually heart failure [123].

In diabetic cardiomyopathy, the balance between glucose and FA utilization shifts to the FA side. As a crucial step, CD36 permanently relocates to the sarcolemma, which increases FA uptake [124]. The increased FA uptake is beyond the mitochondrial β-oxidation capacity, initiating cellular lipid accumulation. The accumulated lipids activate the serine/threonine cascade, which subsequently inhibits GLUT4 translocation and insulin receptor substrate 1 (IRS1), leading to lipid-induced insulin resistance and finally cardiac dysfunction [125,126].

## 4. Interplay between STIM1 and Glucose Metabolism

### 4.1. STIM1 Is Involved in the Regulation of Glucose Metabolism

There is a massive number of findings relating STIM1 to glucose metabolism in different organs or cell types. In non-excitable cells, research showed that STIM1-mediated SOCE contributes to glucose uptake via the insulin pathway, Ca^2+^–calcineurin pathway or regulating the expression of GLUT [113,127,128]. In addition, Johnson et al. showed that STIM1 knockdown decreased glycolysis [129]. Likewise, Zhao et al. observed that STIM1 deficiency shifted glycolysis toward AMPK (a key regulator of both glucose and FA metabolism)-activated fatty acid oxidation by decreasing the expression of key proteins (i.e., GLUT2, GLUT3) involved in glucose uptake and increasing the expression of key proteins related to fatty acid oxidation [130]. Both effects indicate that STIM1 is a positive regulator of glycolysis. The situation is a bit complicated in excitable cells and tumor cells. Gross et al. showed that in melanoma UV-induced SOCE suppression increased glucose uptake [131]. Wilson et al. discovered that the disruption of STIM1 in adult skeletal muscles resulted in enhanced glycolysis upon insulin stimulation [132], and Qiu et al. discovered that the neuron-restricted deletion of STIM1 in female mice increased glucose tolerance with high-fat dieting [133]. Different from non-excitable cells, these findings implicate that STIM1 negatively regulates glucose uptake and glycolysis. However, Olianas et al. showed that STIM1 knockdown inhibited Gq/11-coupled muscarinic acetylcholine receptors-induced glucose uptake in human SH-SY5Y neuroblastoma cells [134], and they also found that in human SH-SY5Y neuroblastoma cells, STIM1-mediated SOCE stimulated AMPK by phosphorylation at Thr 172, and it also increased glucose uptake and the membrane expression of the GLUT1 [135], revealing a positive role of STIM1 in glucose uptake via AMPK phosphorylation and the regulation of GLUT expression. In addition to STIM1, STIM2 was also reported to be involved in Ca^2+^-mediated AMPK phosphorylation at Thr172 [136]. Interestingly, Stein et al. discovered that AMPK downregulated SOCE by the phosphorylation of STIM1/2 in the liver [137], and Wilson et al. found that STIM1 deficiency caused an increase in phosphorylated AMPK at Thr 172 (p-AMPK) in adult skeletal muscle [132]. Although these findings were not obtained in the same types of cells, they still strongly suggested that there is a feedback loop between AMPK-involved energy metabolism and STIM1/2-mediated SOCE. Yet, the evidence provided above shows that STIM1 plays a key role in the regulation of glucose metabolism including glucose uptake and glycolysis in different cell types. Moreover, Collins et al. demonstrated for the first time that the cardiomyocyte-specific deletion of STIM1 downregulated glucose utilization, and the expression of GLUT4 and p-AMPK were reduced [32], which indicates a positive role of STIM1 in the regulation of glucose metabolism in cardiomyocytes.

### 4.2. Glucose Metabolism Influences the Expression and Modification of STIM1

Kono et al. described a specific reduction in STIM1 expression level in type 2-diabetic islets where insulin-induced glucose metabolism was impaired [138]. Sabourin et al. observed that prolonged exposure to supraphysiological glucose concentration impaired SOCE without the changes of STIM1 and Orai1 expression levels in pancreatic β-cells [139]. Tian et al. showed that high glucose decreased the accumulation of STIM1 puncta in the subplasmalemmal ER in islet β cells [140]. Thus, in islets, impaired insulin-induced glucose uptake is related to reduced STIM1 expression, and high glucose could downregulate SOCE by decreasing the translocation of STIM1 to the plasma membrane.

Nevertheless, Edwards et al. reported that STIM1 expression and SOCE were increased in coronary smooth muscle with metabolic syndrome, which showed impaired glucose tolerance and insulin resistance [141]. Similarly, increased STIM1 expression and SOCE have been reported in type 2-diabetic platelets with peripheral artery disease [142]. Moreover, Rong Ma’s lab demonstrated that high glucose upregulated SOCE by increasing STIM1/Orai1 expressions in mesangial cells, and high glucose increased STIM1 expression levels by impairing HNF4α binding activity to the STIM1 promoter [143,144]. Similar findings were also shown in human aortic smooth muscle cells, vascular endothelial cells, neurons, small intestinal smooth muscle cells and islet microvascular endothelial cells [145,146,147,148,149]. In summary, opposite to the situation in islets, impaired insulin-induced glucose uptake is associated with increased STIM1 expression, and high glucose could increase SOCE by increasing STIM1 expression in other kinds of cells.

Although it is not so clear if excess glucose would induce glycosylation, some reports showed that increased glucose uptake seems associated with increased N-glycosylation [150,151]. Interestingly, it was reported that the N-glycosylation of STIM1 may decrease its Ca^2+^ binding affinity, and mutations at the N-glycosylation sites could profoundly affect SOCE [152,153]. In addition to glucose itself, some by-products (e.g., methylglyoxal, pyruvic acid) of glucose metabolism could modify STIM1 as well. Methylglyoxal could spur the production of oxidized glutathione, which leads to the S-glutathionylation of STIM1, increasing SOCE [95,154]. Pyruvic acid, a by-product of glycolysis and also a critical rate-limiting substrate for mitochondrial respiration, increased SOCE by reducing the Ca^2+^-dependent inactivation of STIM1–Orai1 channels [155]. Taken together, it implies that glucose metabolism influences STIM1-mediated SOCE by regulating the expression and post-translational modification of STIM1.

### 4.3. STIM1 and Altered Glucose Metabolism in Cardiac Hypertrophy (and Diabetic Cardiomyopathy)

The increased expression of STIM1 has been reported in cardiac hypertrophy [16,17,18,19,21,22], while the decreased expression of STIM1 has been shown in diabetic cardiomyopathy [30]. Of note, the metabolic changes caused by STIM1 deficiency in cardiomyocytes are similar to those in a heart suffering from diabetic cardiomyopathy [32]. Those findings are suggestive of a notion that STIM1 acts as a “metabolic decider” which can determine cardiac energy substrate preference. In addition, STIM1L has been reported to express in hypertrophic adult cardiomyocytes [24], and STIM1A/β found in several tissues including the heart has been shown to have its splice-specific partners [45], suggesting that a splicing shift of STIM1 is also involved as a “metabolic decider”. Based on the aforementioned evidence of the interplay between STIM1 and glucose metabolism, we propose that STIM1 mediates the glucose preference in cardiac hypertrophy. In addition, the relationship between STIM1 and altered glucose metabolism in diabetic cardiomyopathy will also be discussed.

#### 4.3.1. STIM1 and Glucose Uptake

Reports show that STIM1 may promote glucose uptake via its actions on several pathways. STIM1 has been shown to positively modulate the expression levels of GLUT1, GLUT2, and GLUT3 in several types of cells [113,129,130,135]. In cardiomyocytes, STIM1 deficiency only significantly decreased protein levels of GLUT4 [32], which is one of the two major mediators (GLUT1 and GLUT4) of glucose uptake into cardiomyocytes [156,157]. Therefore, increased STIM1 expression in cardiac hypertrophy could enhance glucose uptake through upregulating GLUT4 (Figure 4➀). Although it is believed that decreased glucose consumption is a result of lipid accumulation in diabetic cardiomyopathy [125,126], the reduction in STIM1 could also contribute to the decrease in glucose utilization by downregulating GLUT4 expression in diabetic cardiomyopathy.

Moreover, STIM1 deletion in cardiomyocytes decreased insulin-induced AKT phosphorylation at Ser473 and increased AS160 phosphorylation which is crucial for insulin-induced glucose uptake [32,158]. In addition, exercise can improve insulin resistance [159]; however, STIM1 ablation in cardiomyocytes reduced exercise-induced AKT phosphorylation at Ser473 [160]. Together with the fact that reduced STIM1 expression is also associated with impaired insulin-induced glucose uptake in other types of cells [127,138], we guess that decreased STIM1 might contribute to impaired glucose uptake by inhibiting the insulin pathway in diabetic cardiomyopathy, while increased STIM1 may upregulate insulin-induced glucose uptake via enhancing AKT phosphorylation in cardiac hypertrophy (Figure 4➁).

In addition, AMPK phosphorylation at Thr172, which is important for glucose uptake, is markedly decreased in hearts from cardiomyocyte-restricted STIM1–KO mice [32]. Consequently, a lower STIM1 expression in diabetic cardiomyopathy could be responsible for the impaired glucose uptake by diminishing AMPK phosphorylation at Thr172, while a higher STIM1 expression in cardiac hypertrophy could increase AMPK phosphorylation at Thr172 to enhance glucose uptake (Figure 4➂). Meanwhile, as a feedback, the alterations of phosphorylated AMPK could regulate STIM1-mediated SOCE by influencing STIM1 phosphorylation [161] (Figure 4➄), but the effects seem not strong enough to reverse markedly increased/reduced STIM1 in cardiac hypertrophy/diabetic cardiomyopathy.

Nevertheless, it is difficult to distinguish if these changes of the key proteins for glucose utilization in diabetic cardiomyopathy are from a direct effect of STIM1 deficiency or a secondary effect of lipid accumulation caused by STIM1 deficiency. Further investigations are needed to provide more information.

#### 4.3.2. STIM1 and Glycolysis

Consistent with reports showing that STIM1 and STIM1-mediated SOCE are positively involved in glycolysis in several different cells [113,129,130], total glycolysis is significantly decreased in hearts from cardiomyocyte-restricted STIM–KO mice [32]. It is possible that these effects might be achieved by the alterations of the expression and activity of enzymes related to glycolysis. Indeed, AMPK has been found to phosphorylate and activate phosphofructokinase-2 (PFK-2), which participates in the production of fructose-2,6-bisphosphate, which is a potent activator of a key enzyme in glycolysis, PFK-1 [162]. Therefore, enhanced AMPK activation caused by increased STIM1 expression could activate more PFK-1 to elevate glycolysis in cardiac hypertrophy (Figure 4➃), while the case is another way around in diabetic cardiomyopathy. However, there is also a paradoxical result showing that PFK-1 is increased in hearts from cardiomyocyte-restricted STIM-KO mice [32]. Therefore, the effects of altered STIM1 level on the expressions or activity of key enzymes responsible for glycolysis still need to be determined.

#### 4.3.3. STIM1 and Glucose Oxidation

Glucose oxidation is decreased in diabetic cardiomyopathy and uncoupled with increased glycolysis in cardiac hypertrophy [7]. A recent report showed that glucose contributed to an increased biosynthesis of aspartate and increased biomass during phenylephrine-induced cardiac hypertrophy [163]; thus, it is possible that increases in other energy substrate metabolisms (e.g., lactate metabolism, amino acid metabolism) could direct excessive pyruvate from upregulated glycolysis away from oxidation in mitochondria (Figure 4➅). Glucose oxidation was shown to be decreased in hearts from cardiomyocyte-restricted STIM1-KO mice, which was accompanied by the decreased activity of pyruvate dehydrogenase (PDH) [32]. PDH is the key enzyme in glucose oxidation, which promotes irreversible pyruvate decarboxylation to acetyl-CoA. PDH could be activated by dephosphorylation via a specific Ca^2+^-sensitive PDH phosphatase [164], and it is inhibited by phosphorylation with specific PDH kinase (PDK)-like PDK4 [165,166]. PDK4 is the major PDK in the heart, which is inhibited by pyruvate [166]. Results showed that the PDK4-dependent decrease in glucose oxidation in diabetic cardiomyopathy is likely due to lipid accumulation instead of lower STIM1 expression [32]. Similarly, pyruvate transport into the mitochondrial matrix [118], PDH activity and resulting glucose oxidation showed no sign of decreases in hypertrophied hearts [117,118,119,120]. Therefore, there seems to be no correlation between enhanced STIM1 expression and pathways involved in glucose oxidation in cardiac hypertrophy.

#### 4.3.4. Feedback from Altered Glucose Metabolism to STIM1

Given that there is an increase in glucose uptake and glycolysis in cardiac hypertrophy [120], glucose itself and some by-products (e.g., methylglyoxal, pyruvic acid) from glucose metabolism are able to modify STIM1 (e.g., N-glycosylation) and increase SOCE [95,152,154,155] (Figure 4➆); therefore, STIM1-mediated SOCE could be further enhanced, accelerating the development of cardiac hypertrophy. Conversely, increased glucose uptake in the hypertrophic heart also leads to glucose-mediated post-translational modifications of proteins (e.g., O-GlcNAcylation) [167,168], and it is observed that O-GlcNAcylation of STIM1 attenuates SOCE in neonatal cardiomyocytes [169], which could be a feedback pathway to downregulate “overactivated” STIM1 in cardiac hypertrophy (Figure 4➆), while the situation might be converse in diabetic cardiomyopathy.

## 5. Interplay between STIM1 and FA Metabolism

### 5.1. FA Influences the Expression and Modification of STIM1

Researchers showed that the binding of long-chain fatty acids (e.g., linoleic acid, palmitate) to CD36 (the major FA transporter in cardiomyocytes [170]) induces IP_3_ production, Ca^2+^ release from ER, and the consequent activation of STIM1-mediated SOCE in mouse gustatory cells, pancreatic β-cells and mouse podocytes [171,172,173,174]. In addition, some reports showed that free FA increased STIM1-mediated SOCE in β-TC3 cells and pancreatic MIN6 β cells [175,176]. However, it has also been shown that FA treatment decreased STIM1 protein levels in mouse coronary ECs [30] and inhibited STIM1–Orai1 coupling via perturbing STIM1 oligomerization in COS-7 cells [177]. In a consistent manner, Wilson et al. showed that lipid overload decreased SOCE via protein kinase C without changing the expressions of STIM1 or Orai1 in liver cells, resulting in ER stress and amplified lipid accumulation [178]. In general, similarly with glucose, the effect of FA on STIM1 and STIM1-mediated SOCE is cell type-dependent: FA increases STIM1 and STIM1-mediated SOCE in islets but shows an opposite phenomenon in other cells. Similar to the modification of STIM1 by glucose, it has been found that STIM1 could be S-acylated at cysteine 437 by lipids, which is required for the assembly of STIM1 into puncta with Orai1 and full STIM1–Orai1 channel function [179]. Taken together, these findings indicate that FA influences STIM1-mediated SOCE by regulating the expression and post-translational modification of STIM1.

### 5.2. STIM1 Is Involved in the Regulation of FA Metabolism

Khan’s lab found that STIM1-deficient mice lost the spontaneous gustatory preference for fat [180]. Conversely, Maus et al. observed pathological accumulations of lipid droplets in the liver, heart and skeletal muscle of SOCE-deficient mice. In addition, there was a similar phenotype in fibroblasts from patients with loss-of-function mutations in STIM1 and Orai1. They also showed that SOCE was crucial for regulating the mRNA levels or expression levels of lipases (which are able to hydrolyze lipid droplets to form free fatty acids for β oxidation) and fatty acid β-oxidation-related proteins in fibroblasts and NIH3T3L-1 cells [181]. Moreover, Baumbach et al. found that STIM (an ancestral variant) knockdown in fat storage tissues increased the amount of fat store in *Drosophila* [182]. Most findings indicate that STIM1 has a negative effect on FA utilization. Similarly, the research from Collins et al. showed that the cardiomyocyte-restricted deletion of STIM1 increased lipid accumulation and the expressions of enzymes related to fatty acid synthesis [32], which provides more direct evidence that STIM1 is negatively involved in the regulation of FA utilization in cardiomyocytes.

### 5.3. STIM1 and Altered FA Metabolism in Diabetic Cardiomyopathy (and Cardiac Hypertrophy)

Based on the aforementioned evidence of the interplay between STIM1 and FA metabolism, we speculate that STIM1 may mediate the FA preference in diabetic cardiomyopathy and that there is an association between STIM1 and altered FA metabolism in cardiac hypertrophy.

#### 5.3.1. STIM1 and FA Uptake

FA uptake into cardiomyocytes is mainly mediated by CD36, and increased FA uptake in diabetic cardiomyopathy is due to a permanent relocation of CD36 to the sarcolemma [124]. Phosphorylated AMPK at Thr172 plays a key role in regulating CD36 translocation from endosomes to sarcolemma [126,183]. The alkalinization of endosomes trigged by the disassembly of vacuolar H^+^-ATPase (v-ATPase) could also enhance CD36 relocation to the sarcolemma in diabetic cardiomyocytes. Given that AMPK phosphorylation at Thr172 is markedly reduced in hearts from cardiomyocyte-restricted STIM1–KO mice [32], it seems that STIM1 could not contribute to CD36 translocation through phosphorylated AMPK at Thr172. Instead, endosomal enzyme vacuolar v-ATPase could be a mediator, as the genetic deletion of STIM1 in cardiomyocytes decreased both insulin- and exercise-induced AKT phosphorylation at Ser473 [32,160]. The decrease in AKT activity which was caused by reduced phosphorylation at Ser473 may inhibit the assembly and subsequent activation of v-ATPase via activating its downstream target mTORC1 (mammalian target of rapamycin complex 1) less in cardiomyocytes [36]. Therefore, we propose that instead of influencing CD36 expression, decreased STIM1 expression in diabetic cardiomyopathy could increase FA uptake by the relocation of CD36 to the sarcolemma through the inactivation of the AKT–mTORC1–v-ATPase signaling pathway (Figure 5➀).

Additionally, a recent report showed that glycogen synthase kinase-3α (GSK-3α) mediates the upregulation of transcription of genes related to fatty acid uptake and storage through PPARα phosphorylation at Ser280 in high-fat diet-induced lipotoxic cardiomyopathy [184], which showed similar FA metabolic changes to those in diabetic cardiomyopathy. Consistently, in diabetic patients’ hearts, enhanced GSK-3α activity and decreased STIM1 expression were observed [184]. A similar phenomenon was also seen in mouse dendritic cells [185]. It would be intriguing to test if there is a link between STIM1 and GSK-3α activity (Figure 5➁), which would provide more insights on the mechanisms by which STIM1 regulates FA metabolism in diabetic cardiomyopathy.

#### 5.3.2. STIM1 and FA Oxidation

In terms of how STIM1 is associated with FA oxidation, our speculation is that STIM1 could decrease carnitine-palmitoyl transferase 1 (CPT1) expression or activity, leading to the reduction in FA oxidation in cardiac hypertrophy (Figure 5➂). CPT1 plays a key role in the regulation of FA oxidation through transporting cytoplasmic FA–CoA into mitochondria. CPT1 was significantly increased in the hearts from cardiomyocyte-restricted STIM1-KO mice [32]. This phenomenon indicates that decreased STIM1 in the development of diabetic cardiomyopathy may increase the CPT1 expression and corresponding FA oxidation rate [186,187,188].

In addition to its inhibition of CPT1 expression, STIM1 also could decrease CPT1 activity in the hearts from cardiomyocyte-restricted STIM1-KO mice [32]. This effect is mediated by inhibiting the acetyl-CoA carboxylase (ACC)–malonyl–CoA pathway [32,189]. It has been shown that the activity of ACC was inhibited by enhanced phosphorylation at Ser79 in the hearts from cardiac-specific STIM1-KO mice [32]. This STIM1-dependent phosphorylation of ACC is not mediated by AMPK, but it might be regulated by other kinases such as protein kinase A (PKA) [32,190]. Nevertheless, ACC is a negative regulator of CPT-inhibitor malonyl-CoA; thus, STIM1-dependent inhibition of ACC would decrease the activity of CPT1.

PPAR and PGC-1α, which regulate the expression of genes encoding FA oxidation enzymes, were upregulated in diabetic cardiomyopathy, contributing to enhanced FA oxidation [191]. Meanwhile, in cardiac hypertrophy, decreased FA oxidation was caused by the decreased expression of PPAR and PGC-1α [192,193,194,195,196,197,198]. Even though the genetic deletion of STIM1 in cardiomyocytes had no effect on the expression of PPARα and PGC-1α [32], STIM1 deletion together with STIM2 deletion decreased PPAR and PGC-1α mRNA levels [181]. Thus, it is possible that STIM2, but not STIM1, may regulate PPAR and PGC-1α expression and corresponding changes of FA oxidation (Figure 5➃).

#### 5.3.3. STIM1 and Lipid Accumulation

Lipid accumulation, which is due to an imbalance between FA uptake and FA oxidation, is one of the key pathologic features of diabetic cardiomyopathy that precedes the onset of contractile dysfunction [7,199,200]. Interestingly, triglycerides and lipid droplets were increased in hearts from cardiomyocyte-restricted STIM1-KO mice [32]. Considering that STIM1-mediated SOCE has been shown to positively regulate the mRNA levels or expression levels of lipases (which are able to hydrolyze lipid droplets to form free fatty acids) [181,182] (Figure 5➄), it is not surprising that decreased STIM1 expression in diabetic cardiomyocytes could reduce the hydrolyzation of lipid droplets and thus cause lipid accumulation.

#### 5.3.4. Feedback from FA Metabolism to STIM1

How increased FA influences STIM1 expression in diabetic cardiomyopathy still needs to be determined. Based on observations from Wilson et al. [178], a reasonable hypothesis could be that lipid overload in diabetic hearts may decrease STIM1-mediated SOCE via protein kinase C (Figure 5➅), leading to ER stress and in turn amplified lipid accumulation. In addition, STIM1 can be modified by lipids, for example being S-acylated at cysteine 437 [179]. Therefore, increased lipid accumulation in diabetic cardiomyopathy could cause more S-acylation of STIM1 and thus increase SOCE to some extent, acting as feedback to rescue decreased STIM1 expression-induced SOCE reduction (Figure 5➅).

## 6. Conclusions and Future Perspectives

In this review, we collected evidence to show and discuss the emerging role of STIM1 in the regulation of cardiac energy substrate preference in cardiac metabolic diseases and speculated the possible underlying mechanisms. In cardiac hypertrophy, increased STIM1 expression could induce the energy substrate preference toward glucose through regulating glucose transporters, key kinases and enzymes of energy metabolism and mitochondrial function. Conversely, in diabetic cardiomyopathy, decreased STIM1 levels could induce FA preference by regulating FA transporters, key kinases and enzymes of energy metabolism and mitochondrial function in a way opposite to cardiac hypertrophy. In conclusion, it seems that STIM1 could work as a “decision-maker” of cardiac energy substrate preference.

Supporting our speculations, a growing number of studies have shown the role of STIM1 as a potential target to rescue cardiac metabolic diseases. For example, the suppression of STIM1-medaited SOCE, whatever the methods applied (e.g., pharmacological inhibition, genetic inhibition, inhibition through STIM1 regulators), all alleviated the development of cardiac hypertrophy [16,17,24,201,202,203,204,205,206,207,208]. In contrast to in cardiac hypertrophy, STIM1 expression is reduced in diabetic cardiomyopathy, which makes it difficult to find a suitable method to improve STIM1 activation. Thus, not much can be found yet from the literatures using STIM1 as a target to prevent diabetic cardiomyopathy. Inhibiting STIM1 negative regulators could be a way, and identifications of STIM1 negative regulators would be helpful. Notably, the over-inhibition of STIM1-mediated SOCE could induce heart failure (by causing diabetic cardiomyopathy?) [32,158], which should be taken into account when preventing cardiac hypertrophy via inhibiting STIM1-mediated SOCE.

STIM1 has been reported to interact with a number of proteins including TRPC, Ca_v_1.2, SARAF, EB1, Gelsolin, STING, POST, phospholamban, SERCAs, and STIMATE [52,57,78,79,80,81,82,83,84,85]. It is interesting to note that besides being regulators of STIM1-mediated SOCE signaling, some of the STIM1 partners have other important cellular functions. For example, STIM1-Gelsolin contributes to cytoskeletal regulation in neonatal cardiomyocytes [84]. To better understand the role of STIM1 in the regulation of cardiac metabolism, identifying more STIM1 partners and underlying their functions are still highly needed. In addition, angiotensin-converting enzyme (ACE) inhibitors are established in the treatment of cardiovascular diseases [209]. Patients diagnosed with diabetic cardiomyopathy and pressure overload-induced cardiac hypertrophy often receive treatment with ACE inhibitors [210]. It would be interesting to investigate the potential interactions between the ACE inhibitors’ metabolic actions and STIM1 signalization in the regulation of cardiac energy substrate preference. Moreover, STIM1 could also be a potential cardioprotective target for the patients who suffer cardiotoxicity due to various anti-cancer therapies; future investigations are needed though.

In summary, it is tempting to see that the crosstalk between STIM1 and energy substrate preference in cardiac metabolic diseases is very complex. The precise causal relationship between them still awaits further elucidation. And this review may provide some possible directions for future studies to explore the precise mechanisms by which STIM1 works as a “balancer” to regulate cardiac energy metabolism.

## Figures and Tables

**Figure 1 ijms-24-13188-f001:**
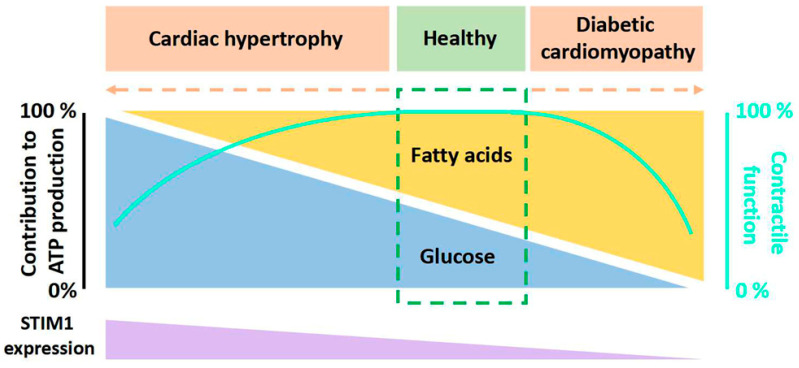
Diagram of the relation between the relative contributions of fatty acids and glucose to overall myocardial ATP production, contractile function and STIM1 expression. Contributions from lactate, ketone bodies, and amino acids are not shown here. In a healthy heart, there is a balance between fatty acids utilization and glucose utilization to produce ATP. However, when cardiac metabolic diseases like cardiac hypertrophy and diabetic cardiomyopathy occur, this balance shifts toward the predominant utilization of a single substrate, either fatty acids or glucose. This altered substrate preference is associated with changes in STIM1 expression.

**Figure 2 ijms-24-13188-f002:**
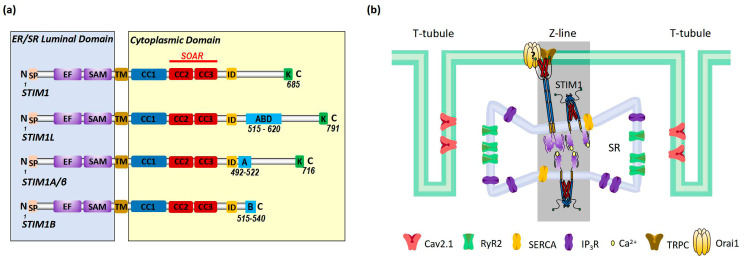
Schematic presentation of (**a**) domains of STIM1 and its alternative splicing variants and (**b**) the localization of STIM1 in cardiomyocytes. (**a**) Cartoon of domains of STIM1 and alternative splicing variants. STIM1 is composed of the ER luminal domains that contain signal peptide (SP), the EF hand and SAM domain, transmembrane (TM) domain and the cytosolic domains that contain CC1, CC2, and CC3, the inhibitory domain (ID) and polybasic domains (K). The EF-SAM domain is able to bind Ca^2+^. CC2 and CC3 domains form the minimum domain of STIM1 to activate Orai1 (STIM1-Orai1 activating region, SOAR). Compared to STIM1, STIM1L has an extra actin-binding domain (ABD) of 106 amino acids. STIM1A/β has an extra 31 amino acids (A) after ID. STIM1B lacks 170 amino acids in the cytoplasmic domain but has an additional domain of 26 amino acids (B). (**b**) Cartoon illustration of the localization of STIM1 in cardiomyocytes. STIM1 and Orai1 are resident in SR at the Z-lines and sarcolemma, respectively. Close to STIM1 and Orai1, other key proteins involved in Ca^2+^ handling include the ryanodine receptor 2 (RyR2), voltage-gated Ca^2+^ channel Ca_v_2.1, SR Ca^2+^ ATPase (SERCA) and inositol-1,4,5-triphosphate receptor (IP_3_R).

**Figure 3 ijms-24-13188-f003:**
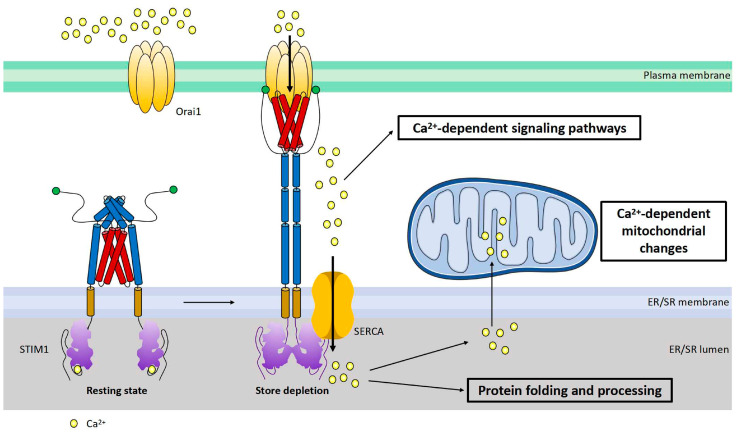
Cartoon illustration of activation and function of STIM1. STIM1 is a single transmembrane protein resident in ER/SR. Orai1 channel functions as a hexamer on plasma membrane. Under the resting state, the cytosolic region of STIM1 maintains an inactive folding configuration. Upon SR Ca^2+^ store depletion, the loss of Ca^2+^ triggers the dimerization of the STIM1 EF-SAM domain in the ER lumen, leading to the uncaging of its SOAR domain in the cytosol. SOAR would then bind and open Orai1, the pore-forming protein, inducing store-operated Ca^2+^ entry (SOCE). In addition to its critical roles in mediating many Ca^2+^-dependent signaling pathways, SOCE is also crucial for the maintenance of Ca^2+^ homeostasis within the ER/SR and mitochondria. Proper-sized ER Ca^2+^ store is essential for correct protein folding and processing as well as for maintaining Ca^2+^-dependent mitochondrial changes of the cell.

**Figure 4 ijms-24-13188-f004:**
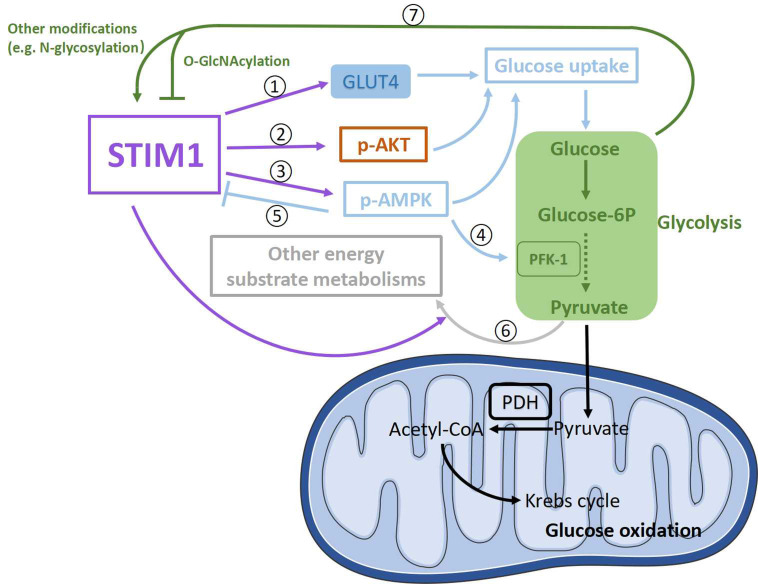
Schematic presentation of possible signaling pathways involved in the crosstalk between STIM1 and glucose metabolism in cardiomyocytes. STIM1 is positively involved in glucose uptake through upregulating GLUT4 expression levels ➀ and the phosphorylation of AKT (p-AKT) ➁ and AMPK (p-AMPK) ➂. P-AMPK could upregulate glycolysis through enhancing PFK-1 activity ➃. In addition, p-AMPK could negatively regulate STIM1 via phosphorylating STIM1 ➄. Moreover, STIM1 might downregulate glucose oxidation through increasing other energy substrate metabolisms to drive pyruvate away from oxidation ➅. In turn, by-products from glucose metabolism could alter STIM1 activity by modifying STIM1 ➆.

**Figure 5 ijms-24-13188-f005:**
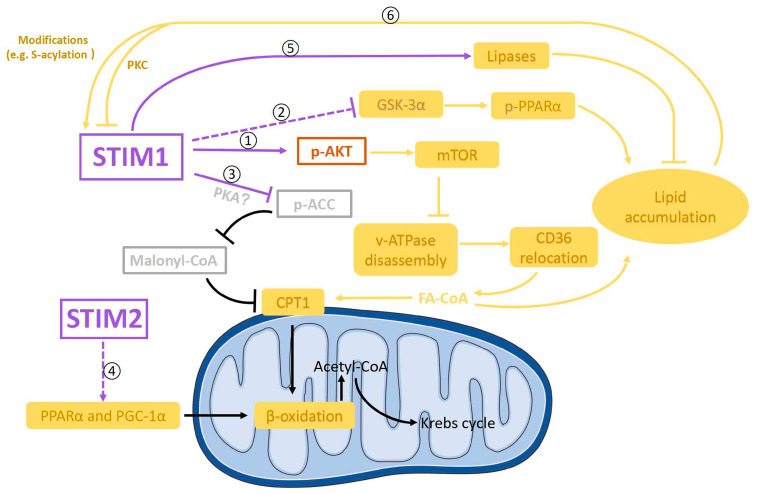
Diagram showing possible signaling pathways involved in the crosstalk between STIM1 and FA metabolism in cardiomyocytes. STIM1 is negatively involved in fatty acids (FA) uptake. STIM1 could upregulate the phosphorylation of AKT (p-AKT) and thereby downregulate FA uptake through CD36 relocation to sarcolemma via the m-TOR–v-ATPase axis ➀. In addition, STIM1 might also negatively regulate lipid accumulation through inhibiting the GSKα-3–PPARα pathway ➁. Moreover, STIM1 could downregulate the phosphorylation of ACC probably through PKA, which could increase FA oxidation through the ACC–malonyl–CoA–CPT1 axis ➂. STIM2 might be positively involved in FA oxidation by upregulating peroxisome proliferator-activated receptor α (PPARα) and PPARγ coactivator-1α (PGC-1α) ➃. Furthermore, STIM1 could reduce lipid accumulation by activating lipases, resulting in the hydrolyzation of lipid droplets into free FA ➄. In turn, by-products from FA metabolism could increase STIM1 activity by modifying STIM1 ➅.

## Data Availability

Not applicable.

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
