# Peer review of "Role of STIM1 in the Regulation of Cardiac Energy Substrate Preference"

_ijms, 2023, doi:10.3390/ijms241713188_

Round 1

Reviewer 1 Report

Overall the review by Liu et al is a very interesting summary on a potential role of STIM1 for the regulation of energy substrates in the heart. It is a very thorough review and considers a large number of published references. Although the review focusses on STIM1, it is important that the authors mention that STIM2 may also have an important role in guiding metabolism, especially as it regulated basal Calcium. Given the considerable number of references, a quick literature search on “STIM2” and “Cardiac” or “heart” reveals a relatively small number of hits (21) and the authors may want to consider to discuss some of those.

Line 60: mention role of STIM2 as rheostat for basal cytosolic Calcium concentrations (Brandmann, T Meyer, Cell 2007)

Line 61: rephrase: Sensing ER/SR store depletion, STIM1 and STIM2 get activated and engage with the pore forming Orai1 ion channel proteins, localized in the plasma membrane, thereby triggering channel opening and activating the previously characterized ICRAC (Ca2+ release activated currents) (ref Hoth and Penner 1992). As most investigators use Ca2+ imaging techniques to report STIM-ORAI activity, this process is usually termed Store-operated Calcium Entry (SOCE).

Line 79:, the expression of STIM1 has been shown to decrease in….

Line 95: Would cite additional  reviews on STIM function (see Soboloff, Nat Rev 2012; Prakryia and Lewis 2015)

Line 128: There are at least three  (databases predict more) alternative splice variants …

Line 130 cite [Knapp et al] for expression in astrocyte, heart, kidney and testis (first paper out – should be 41)

Line 131 and ff:  since there is some confusion with splice variant naming, I would recommend always using both: ie: While STIM1A/ß has been show to more efficiently mediate..  [41- should be 42], it was shown to reduce SOCE in astrocytes in heterologous expression, while at the same time increasing NFAT translocation due to a more efficient recruitment of the NFAT signalosome [42-should be 41]. Indeed, it is possible that differential NFAT activation and not small differences in SOCE mediate a splice variant specific effect in cells or else that the functions of STIM1A/ß may differ in different cell types.

Line 141: in non-stimulated cells, STIM1 is distributed throughout the ER.

Take out unexcitable (see also line 269), there is no unexcitable cell. Replace with cells at rest or unstimulated cells. Or do you mean electrically non-excitable? Define in the beginning what is meant by the term.

Does the SR at Z lines contain less Ca? Why is STIM1 localized only there?  Make this more clear in the text, combine with line 151.

156: Direct activation of TRPC channels by STIM1 is controversial. Discuss in more detail. Are Orai2 and/or Orai3 expressed in the heart?

164 Reference 41 format.

Line 176: mention SNP in rat STIM1 which causes spontaneous hypertension (but cannot find reference anymore)

Line 195 (see ref Knapp et al for additional splice-specific interaction partners of STIM1)

Line 230: Moreover, a study in T cells has shown…  (Reference is missing!)

273: is this clearly STIM1 or SOCE in general?  There is evidence that STIM2 is the preferential activator of AMPK (see Chauhan et al 2019)

Line 291 ff: Inconsistent!  Sabourin observed no changes in STIM1, ORAI1 expression but then consistently Tian shows decreased expression?  Please be clear whether you talk about expression (mRNA), protein levels or protein localization.

Note that Tian et al utilized overexpressed tagged STIM1 which may show differential effects when compared to endogenous STIM1 used in the below mentioned studies.

309 ff:  Glucose cannot directly induce N-glycosylation and see also earlier report by Kilch et al on the differential role of STIM1 N-glycosylation on SOCE (Kilch et al., JBC 2013) and on Orai1 (Dörr et al, Sci Sign 2016). The study by Choi et al used artificial crosslinking of glucose to Stim1 which is different from naturally occurring  transfer of mannose rich glycan trees by the oligosaccharyltranferases within the ER.

Line 322: Reference 29 describes STIM1 levels in platelets, not in the heart!

Line 324 and ff:  Carefully discuss a potential differential role of STIM1 splice variants. It is possible that a splicing shift is also involved as “metabolic deciders”.

Line 333: sentence incomplete: “While” indicates that an opposing statement follows. Check also report by Maus and Feske on STIM meditated metabolism in T cells.

Figure 4 is a bit too speculative and contains typos. Is there any evidence that O-Glycosylation inhibits STIM1? Also although a decrease in glucose may induce downregulation of glycosylation, it is not so clear if excess glucose induces glycosylation.

Also, the figure lacks the main driving force for mitochondrial metabolism, likely the STIM1 mediated influx of Ca driving Calcium uptake into the mitochondria either directly though cytosolic Ca or indirectly after store refilling by the transfer of ER calcium to mitochondria.

Line 448: Cave: Drosophila only has one STIM gene, not STIM1 or STIM2, but ancestral variant.

Line 467: But this is not logical: If Phosphorylation at Thr172 is important for CD36 translocation to the Sarcolemma, and this is reduced in STIM1KO mice, then STIM1 should have an effect on CD36 translocation?

Line 495: see also paper by Schmid et al , Plos One, 2014 on SOCE and GSK-3.

Some minor edits are necessary.

Reviewer 2 Report

This is an interesting review regarding the role of STIM1 in the regulation of cardiac energy substrate preference. This paper reviews some future directions for research studies to investigate the exact mechanisms by which STIM1 might regulate cardiac energy metabolism, including very common cardiac metabolic diseases (e.g., T2DM and HTN).

Some minor suggestions are presented below.

In the title: ‘Role of STIM1 in the regulation of cardiac energy substrate preference?’ the ‘?’  [at the end] is not needed.

Perhaps, in the ‘Conclusions and future perspectives’ section, the Authors may briefly address a common clinical scenario, in which many patients diagnosed with T2DM (e.g., diabetic cardiomyopathy) and HTN (e.g., pressure-overloaded hypertrophic heart) receive treatment with ACEI. In particular, it would be interesting to know what are the potential interactions between the ACEI’s metabolic actions and STIM1 signalization in the regulation of cardiac energy substrate preference.

Also, the Authors may comment on the issue of whether, in the future, STIM1 could represent a potential innovative target in cardio-oncology, especially for patients, who suffer from cardiotoxicity related to various anticancer therapies.
